# Vitamin D deficiency and severity of pneumonia in Indonesian children

**Vicka Oktaria**[ID][1,2,3,4]*, **Rina Triasih**[4,5], **Stephen M. Graham**[1,2], **Julie E. Bines**[1,2], **Yati Soenarto**[4,5], **Michael W. Clarke**[6], **Mike Lauda**[5], **Margaret Danchin**[1,2]

1 Faculty Medicine, Department of Paediatrics, Dentistry and Health Sciences, The University of Melbourne, Melbourne, Australia, 2 Murdoch Children's Research Institute, Royal Children's Hospital, Melbourne, Australia, 3 Faculty of Medicine, Department of Biostatistics, Epidemiology and Population Health, Public Health and Nursing, Universitas Gadjah Mada, Yogyakarta, Indonesia, 4 Faculty of Medicine, Center for Child Health–Pediatric Research Office (CCH_PRO), Public Health and Nursing, Universitas Gadjah Mada, Yogyakarta, Indonesia, 5 Faculty of Medicine, Child Health Department, Public Health, and Nursing, Universitas Gadjah Mada/Dr. Sardjito Hospital, Yogyakarta, Indonesia, 6 Faculty of Health, and Medical Sciences, Metabolomics Australia, Centre for Microscopy, Characterisation, and Analysis, and School of Biomedical Sciences, The University of Western Australia, Perth, WA, Australia

* vicka.oktaria@ugm.ac.id

## Abstract

### Objective

To determine the prevalence of vitamin D deficiency in Indonesian children hospitalized with pneumonia and evaluate the association between vitamin D status and severity of pneumonia.

### Methods

A hospital-based cross-sectional study was conducted from February 2016 to July 2017 in two district hospitals in Yogyakarta province, Indonesia. Infants and young children aged 2–59 months hospitalized with pneumonia were recruited. Serum blood samples were collected on admission and analyzed for total serum 25-hydroxyvitamin D3 and 25-hydroxyvitamin D2 concentrations using liquid chromatography-tandem mass spectrometry. Vitamin D deficiency was defined as a level of serum vitamin D <50 nmol/L. The association between vitamin D deficiency and severity of hospitalized pneumonia according to WHO criteria, including the presence of danger signs, hypoxemia (SpO2 in air below 90%), duration of hospitalization, and admission to Intensive Care Unit (ICU), was analyzed using logistic regression.

### Results

133 children with WHO-defined pneumonia were enrolled in the study and 127 (96%) had their vitamin D status determined. The mean vitamin D concentration was 67 (± 24 SD) nmol/L and 19% of participants were vitamin D deficient. Age younger than 6 months was associated with prolonged hospitalization (> 5 days) and low birth weight and poor nutritional status on admission were risk factors for hypoxemia. However, vitamin D status was not associated with the presence of danger signs, duration of hospitalization, or hypoxemia.

**Data Availability Statement:** All relevant data are within the manuscript.

**Funding:** This study was supported by Murdoch Children's Research Institute in the form of funding awarded to SMG, Schlumberger foundation faculty

for the future in the form of funding awarded to VO,
Indonesia Endowment Fund for Education (LPDP)
Ministry of Finance in the form of a grant awarded
to VO (20130822080370), the David Bickart
Clinician Research Fellowship from the University
of Melbourne awarded to MD, Australia-Indonesia
Centre (AIC) in the form of a grant awarded to MD
and YS (01HSP1MELDancUGM003), and
infrastructure funding from the Western Australian
State Government, in partnership with the
Australian Federal Government, through
Bioplatforms Australia and the National
Collaborative Research Infrastructure Strategy
awarded to MWC. MCRI is supported by the
Victorian Government Infrastructure Fund. The
funders had no role in study design, data collection
and analysis, decision to publish, or preparation of
the manuscript.

**Competing interests:** The authors have declared
that no competing interests exist.

## Conclusions

One in every five children hospitalized with pneumonia was vitamin D deficient. Vitamin D status was not associated with the severity of pneumonia.

## Introduction

Micronutrient supplementation in children has an established role in public health and clinical management of pneumonia. The benefits of vitamin A supplementation, for example, have long been recognized for prevention of morbidity and mortality in young children (<5 years), and for improving clinical outcome for measles pneumonia [1,2]. Therefore, vitamin A supplementation is part of the Global Action Plan for Pneumonia and Diarrhea framework [3]. Zinc supplementation has also been shown in populations at risk of zinc deficiency to improve outcomes in children with diarrhea or pneumonia [4]. More recently, vitamin D supplementation has been considered as a potential strategy for the prevention and treatment of pneumonia [5,6] but evidence from countries with high burden of pneumonia are limited.

Pneumonia and vitamin D deficiency (defined as serum vitamin D level < 50 nmol/L) are common in Indonesian children [7,8]. Indonesia was ranked seventh among countries globally for the number of child pneumonia deaths by UNICEF in 2015 [3]. Community acquired pneumonia accounted for 18% of all hospitalizations in Indonesian children < 5 years with a case fatality rate between 5% and 20% [9]. Vitamin D deficiency has been reported common in one in every three to five Indonesian children aged between 6 months to 12 years old, and nine in every ten newborns [10,11].

Vitamin D modulates the immune and inflammatory response against infections [12]. A meta-analysis using individual participant data reported that vitamin D protected from all respiratory infections in adults and children [13]. Vitamin D deficiency at birth has also been associated with more frequent episodes of acute respiratory infections non pneumonia in the first year of life of Indonesian infants [11]. However, the potential role of vitamin D in reducing severity of lower respiratory tract infections, and thereby potentially improving outcomes and reducing hospitalizations, was not clear [13]. There was a limited number of vitamin D supplementation trials conducted in children hospitalized with pneumonia and findings were inconsistent [14–16]. Evidence from observational studies as to whether or not vitamin D deficiency in young children with pneumonia is associated with more severe manifestations and clinical course have been contradictory and depends on the markers of severity used [17–19]. Although vitamin D could be a simple and cheap intervention, WHO highlights the need for more clinical research to inform specific recommendation [20].

We aimed to determine the prevalence of vitamin D deficiency in infants and children aged 2 to 59 months hospitalized with pneumonia at two district hospitals in Indonesia, and to evaluate the association between vitamin D status and the severity and outcome of pneumonia.

## Materials and methods

### Study design, setting, and participants

We conducted a hospital-based cross-sectional study in two district hospitals in Yogyakarta province: Kota Yogyakarta hospital (RSUD Kota, an urban hospital) and Kulon Progo hospital (RSUD Wates, a rural hospital) from 1st February 2016 to 31st July 2017. All infants and children aged 2–59 months who were firstly admitted in one of the two study hospitals with pneumonia during the study period, were eligible for recruitment to the study.

## Study procedures

Following written informed consent from the parent, on admission we documented the clinical characteristics of the child, symptoms and signs of pneumonia including the presence of danger signs, nutritional status, and peripheral oxygen saturation using a standard case report form (S1 File). Pneumonia was defined according to the revised WHO criteria of cough with fast breathing or chest indrawing, without danger signs [21]. Severe pneumonia was defined as pneumonia with any danger sign present including hypoxemia, inability to drink and feed, nasal flaring or grunting, persistent vomiting, lethargy, and convulsion [21]. Hypoxemia was defined as SpO2< 90% in room air. We defined prolonged hospitalization as duration of hospitalization of more than 5 days. This was based on the average length of five days hospital stay for pneumonia reported by Center for Disease Control and prevention (CDC) [22] and was in accordance with the minimum days for standard antibiotic therapy course for children hospitalized with pneumonia by WHO [21]. The similar definition has also been used by other study [23]. Undernutrition was defined by WHO definition: underweight for weight-for-age—3 SD to < -2 SD, stunted for height(or length)-for-age—3 SD to < -2 SD, and wasted for weight-for-height(or length) - 3 SD to < -2 SD. Severe undernutrition was defined as: severely underweight for weight-for-age < -3SD, severely stunted for height(or length)-for-age < -3SD, and severely wasted for weight-for-height(or length) <-3SD [24]. There is no consensus on the recommended serum vitamin D cut off for respiratory health. We defined vitamin D deficiency as a serum (or plasma) 25-hydroxyvitamin D (25(OH)D) concentrations < 50 nmol/L. This is the recommended cut off for optimal bone homeostasis [25]. In this paper, we mainly used < 50 nmol as the cut-off for deficiency but different cut off levels (< 75 nmol/L and < 25 nmol/L) were also tested in our analysis.

A parent-interview questionnaire was completed to collect information on the baseline characteristics, including the potential risk factors for severe pneumonia (e.g., birth weight, prematurity, household crowding, childcare attendance, early solid food, indoor pollution) (S1 File). On the day of admission, a complete blood count test and a chest radiograph were performed. An extra specimen of 2.5 mls of venous blood was collected within day 1 to 3 days of hospitalization for a vitamin D assay at the time of routine blood collection. Clinical management (therapy received including antibiotics, bronchodilators and oxygen therapy) and outcomes (presence of danger signs, admission to ICU, duration of stay in hospital, discharge outcome, and final diagnosis) were recorded.

## Study assays and measurement for vitamin D

Sera from venous blood were assayed at Metabolomics Australia, Centre for Microscopy, Characterization, and Analysis, University of Western Australia, Perth. The total serum 25-hydroxyvitamin D3 and 25-hydroxyvitamin D2 concentrations were measured using liquid chromatography-tandem mass spectrometry using an Agilent 6460 2D LC-MS/MS, Singapore. This was the current gold standard for vitamin D measurement, which has shown excellent agreement with CDC's 25(OH)D3 inaugural vitamin D standardization program (r2 = 0.99) [26]. The limit of detection and the limit of quantitation of the assay for 25-hydroxyvitamin D3 is 0.5 nM and 2 nmol/L, consecutively.

## Sample size calculation

We expected at least 25–70 admissions in each of the two hospitals for pneumonia in under five years old children over the one-year recruitment period. Based on a 10% refusal rate, at least 128 participants enabled description of the prevalence of vitamin D in under-five children

admitted for pneumonia to within 8.7% based on an estimated prevalence of 40% (as estimated from community-based studies), based on a two-sided 95% confidence interval.

## Statistical analysis

Data were double entered to REDCap from data collection instruments and exported to STATA version 15 (Stata Corporation, College Station, Texas) for analysis. Continuous variables were presented as mean ±SD for normal distribution or median and interquartile range for skewed variables. Categorical variables were presented as proportions (%). Vitamin D levels were summarized as continuous and categorical variables.

Indicators of the severity of pneumonia for analysis were presence or absence of danger signs, hypoxemia, prolonged hospitalization, and ICU admission. Binomial categorization of vitamin D was used in the univariate and multivariate logistic regression analysis for exploring the association between vitamin D status and the indicators of severity of pneumonia as binary variables. Results were presented as Odds ratios (ORs) with 95% CIs. A p-value of $< 0.05$ was considered statistically significant.

## Ethics

Ethics approvals for the study were obtained through the UGM Medical and Health Research Ethics Committee (MHREC), Yogyakarta, Indonesia (ethic approval number KE/FK/935/EC/2015) and the University of Melbourne Human Research Committee (ethics approval number 1544817).

## Results

From 1st February 2016 to 31st July 2017, there were 110 out of 286 (39%) and 23 out of 77 (30%) of children admitted with a diagnosis of pneumonia in the rural and the urban hospitals, respectively, that were eligible for recruitment and enrolled in the study (**Fig 1**). Of these, 127 (96%) had vitamin D results available for analysis. No mortality was reported but two ICU admissions were documented.

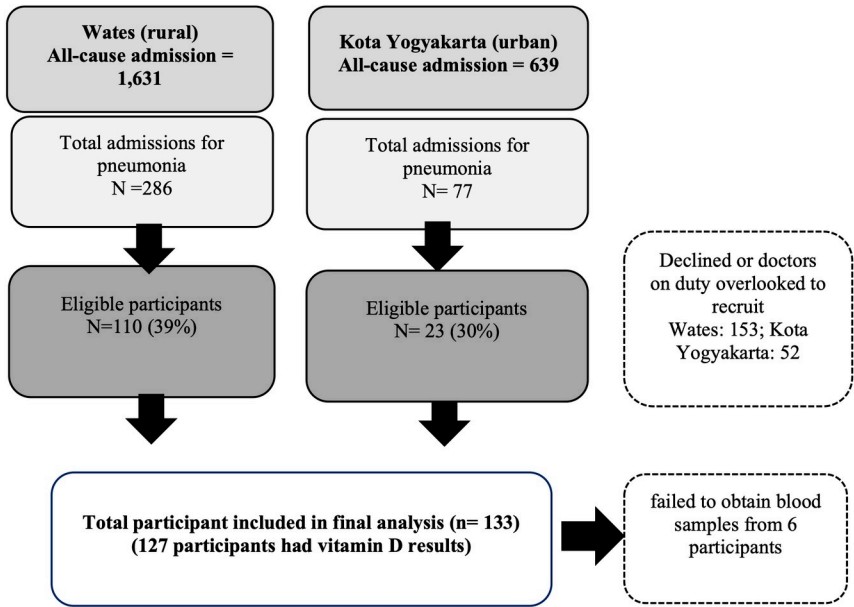

**Fig 1. Hospital study recruitment between the 1st February 2016 to 31th July 2017.**

The median age of participants was 12 (IQR 6–23) months old with the majority of participants from the rural hospital (83%) and of male sex (59%). Sixty-two (47%) of participants were exclusively breastfed until six months and 24 (18%) were aged less than six months at the time of recruitment and were still exclusively breastfed. The mean age of mothers was 32 (SD± 6.4) years and participants were predominantly from low-income families and around one half (52%) had a father who smoked cigarettes (**Table 1**).

Fast breathing was the most common presenting symptoms (95%), followed by fever (78%) (**Table 2**). Chest indrawing was present in 84% of participants and 14% of participants were hypoxemic on admission, although 80% of all recruited participants received oxygen therapy. Thirty participants (23%) were underweight or severely underweight (based on weight for age < -2 SD), and of the 70 participants with weight-for length recorded, 14 (20%) were wasted or severely wasted.

The vitamin D level was normally distributed (**Fig 2**) (mean (± SD): 67 ± 24 nmol/L) with 19% (25/127) of participants having serum vitamin D less than 50 nmol/L. Of these vitamin D-deficient participants, 36% (9/25) had serum vitamin D level less than 30 nmol/L.

Vitamin D deficiency was not associated with the presence of danger signs, the duration of stay in the hospital and hypoxemia (**Table 3**). A sensitivity analysis using a higher and a lower vitamin D cut off (< 75 nmol/L and <25 nmol/L) did not alter the findings (**Table 3**). Participants younger than six months were twice as likely to have prolonged hospitalization compared to older participants (AOR 2.91, 95% CI: 1.23–6.92, **Table 3**). Low birth weight and poor nutritional status on admission were significant independent risk factors for hypoxemia on admission in the hospital.

## Discussion

We found that one in every five children aged 2–59 months admitted to hospital with pneumonia was vitamin D deficient (<50 nmol/L), but vitamin D status was not associated with the severity of pneumonia as indicated by the presence of danger signs, hypoxemia or prolonged hospitalization.

The prevalence of vitamin D deficiency in young children hospitalized with pneumonia in low-middle income countries (LMICs) has been reported to be between 12% to 63% [5,27,28], compared with 9.7% to 30% in high income countries [18,19,29]. Our finding of 19% is similar to that reported in children from Nepal hospitalized with WHO-defined pneumonia (12% of 568 children studied), with comparable demographic profiles but a higher proportion of participants with danger signs or hypoxemia than in our study [17]. A study from Yemen reported a higher prevalence (37%, 29/79) of vitamin D deficiency in children under five years with very severe pneumonia using a lower cut off (< 30 nmol/L), with more than three-quarter of vitamin D deficient participants being clinically rachitic [28]. There are a range of environmental and individual factors that determine vitamin D status that may explain variations of prevalence between settings and populations [30].

Similar to our findings, some observational studies have found that serum vitamin D level on admission was not associated with the severity of pneumonia [19,31,32], while other studies have reported that vitamin D deficiency (< 50 nmol/L) was significantly associated with a higher likelihood for ICU admission, treatment failure, or longer hospital stay [17,18,33]. Studies that reported a significant association had small sample sizes [18,33], did not adjust for confounders [33] or had wide confidence intervals [18], limiting statistical interpretation [33]. While many of our study participants received oxygen therapy, in a subgroup analysis limited to those with hypoxemia, we did not identify an association between vitamin D level and the duration of oxygen therapy received.

**Table 1. Baseline demographics of participants (N = 133)[1].**

| Variables | n (%)[2] |
|---|---|
| Characteristics of participants | |
| Gender (male)—n (%) | 79 (59%) |
| Location- n (%) | |
| Rural (Wates hospital) | 110 (83%) |
| Urban (Kota Yogyakarta hospital) | 23 (17%) |
| Age (in months)—n (%), N = 132 | |
| 2–5 | 32 (24%) |
| 6–11 | 38 (29%) |
| 12–23 | 31 (24%) |
| 24–59 | 31 (24%) |
| Birth weight (in grams), median (IQR), N = 124 | 3000 (2775–3300) |
| Normal (2500–4200 g) | 112 (90%) |
| Low (1500 –<2500 g) | 10 (8%) |
| Very low < 1500 g | 2 (2%) |
| Exclusive breast feeding (EBF)—n (%), N = 111 | |
| No | 25 (23%) |
| EBF until 6 months of age, now completed | 62 (56%) |
| ongoing EBF, in participants currently age ≤ 6 months | 24 (22%) |
| Age started formula feeding (in months), median (IQR), N = 123 | 6 (2–12) |
| Solid food introduced before 6 months of age- n (%), N = 123 | 20 (16%) |
| Socio-environmental characteristics | |
| Mother's age- n (%), N = 123 | |
| < 25 years old | 16 (13%) |
| 25–29 years old | 35 (29%) |
| 30–34 years old | 27 (22%) |
| 35–39 years old | 32 (26%) |
| ≥ 40 years old | 13 (11%) |
| Mother educational level—n (%), n = 126 | |
| Completed middle school or less | 47 (37%) |
| Completed high school | 63 (50%) |
| Attended University | 16 (13%) |
| Family income per month—n (%), n = 126 | |
| IDR < = 1000K | 87 (69%) |
| > IDR 1000 K to < IDR 5000K | 36 (29%) |
| > IDR 5000K | 3 (2%) |
| Solid fuel—n (%) | 62 (47%) |
| Father daily smoker—n (%), n = 126 | |
| No or occasionally | 61 (48%) |
| Yes, daily | 65 (52%) |

1 N = 133 throughout table, unless otherwise specified.

2 variables were presented as n (%) for categorical, mean (±SD) for parametric and median (interquartile range) for non- parametric variables.

1 USD is ~15,000 IDR.

There have been few clinical trials on supplementation of vitamin D for severe pneumonia in hospital settings [15]. Oral vitamin D supplementation 1000–2000 IU/day for 5 days in Indian children (age 2 to 59 months) hospitalized with pneumonia was not associated with the

**Table 2. Clinical characteristics and laboratory profiles of participants (n = 133)[1].**

| Clinical characteristics | n (%)[2] |
|---|---|
| Symptoms and signs—n (%) | |
| Fever | 104 (78%) |
| Cough | 133 (100%) |
| Runny nose | 83 (62%) |
| Wheezing, N = 132 | 74 (56%) |
| Fast breathing, N = 132 | 126 (95%) |
| Chest indrawing, N = 132 | 111 (84%) |
| Stridor | 16 (12%) |
| Any danger signs on admission | 38 (29%) |
| Clinical case definition for pneumonia—n (%) | 133 (100%) |
| Non-severe pneumonia | 95 (71%) |
| Severe pneumonia[3] | 38 (29%) |
| Oxygen saturation on admission (in %), median (IQR), n = 125 | 94 (92–97) |
| Level of oxygen saturation on admission[4] - n (%), N = 125 | |
| ≥ 90% | 107 (81%) |
| < 90% | 18 (14%) |
| Length of stay (in days), median (IQR) | 5 (4–6) |
| Prolonged stay (> 5 days)—n (%) | 49 (37%) |
| Weight for age—n (%) | |
| Normal (-2SD to 2 SD) | 100 (75%) |
| Underweight (- 3 SD to < -2 SD) | 18 (14%) |
| Severely underweight (< -3 SD) | 12 (9%) |
| > 2 SD | 2 (2%) |
| Length for age—n (%), N = 69 | |
| Normal (-2SD to 2 SD) | 42 (61%) |
| Stunted (- 3 SD to < -2 SD) | 17 (25%) |
| Severely stunted (< -3 SD) | 8 (12%) |
| Tall > 2 SD | 2 (3%) |
| Weight for length—n (%), N = 70 | |
| Normal (-2SD to 2 SD) | 51 (73%) |
| Wasted (- 3 SD to < -2 SD) | 9 (13%) |
| Severely Wasted (< -3 SD) | 5 (7%) |
| Overweight > 2 SD | 5 (7%) |
| Anemia—n (%)[5] | 57 (43%) |
| Leukocytosis—n (%)[6] | 105 (79%) |
| Neutrophilia—n (%)[7], N = 132 | 25 (19%) |
| Lymphocytosis—n (%)[8] | 63 (47%) |

[1] N = 133 throughout table, unless otherwise specified.

[2] Variables were presented as N (%) for categorical, mean (±SD) for parametric and median (interquartile range) for non- parametric variables.

[3] severe pneumonia according to WHO 2014 is cough with fast breathing and/or chest indrawing with any danger sign.

[4] hypoxaemia indicated by oxygen saturation < 90%.

[5] Anemia was defined as hemoglobin concentration of < 12.6 gram/dL in age under 6 months, < 12 gram/dL in age between 6 and 23 months, and < 12.5 gram/dL in between 24–59 months.

[6] Leukocytosis was defined as if white blood count > 11.9 x $10^3$ in children aged < 6 mo, > 10.6 x $10^3$ in children aged 6–23 months and >8.5 x $10^3$ in children aged 24–59.

[7] Netrophilia was defined as neutrophil count >70% of the total leucocytes on a peripheral blood smear.

[8] lympocytosis was defined as lymphocytes count >40% of the total leucocytes on a peripheral blood smear.

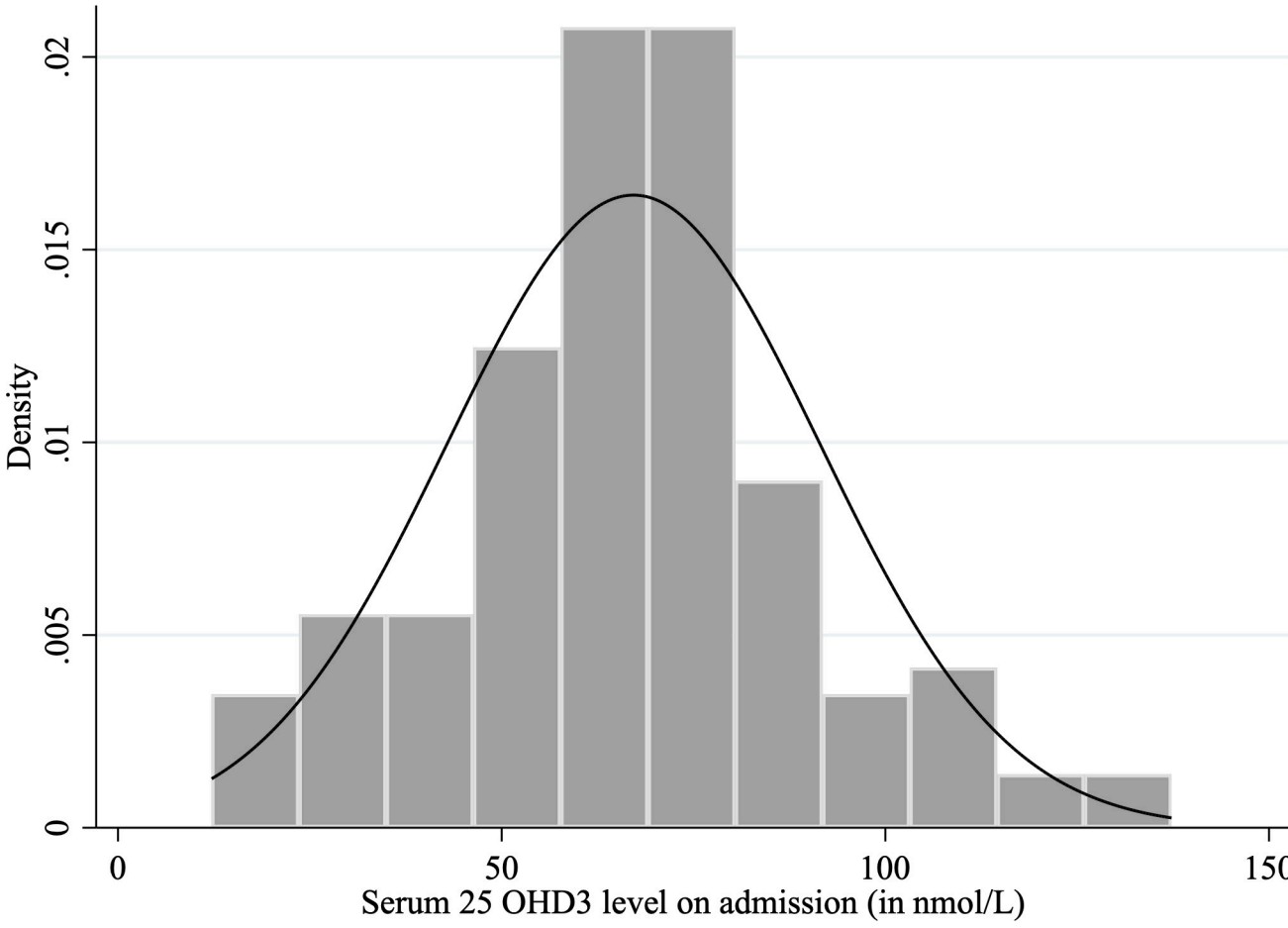

**Fig 2. The distribution of serum vitamin D level on admission.**

resolution time from pneumonia as indicated by the disappearance of chest indrawing and/or danger signs, and hours of hospital stay, but the sample size was small (N = 20) [34]. On the contrary, a higher single dose of oral vitamin D (100,000 IU) was effective to prevent from pneumonia recurrence within 90 days in Afghan children with non-severe pneumonia (N = 453, age 1 to 36 months) but a similar dose of vitamin D supplementation given quarterly in healthy infants in the same population (N = 3,046) did not show prevention from getting pneumonia [35,36]. Different doses of vitamin D tested, and parameters of pneumonia severity have contributed to inconsistent results between trials, and it is difficult to interpret the effect of vitamin D on pneumonia severity based on the assessment of a singular study. A more comprehensive analysis for multiple studies in a Cochrane meta-analysis study of seven randomized controlled trials (N = 1,529 participants) published in 2018, reported that vitamin D supplementation (doses 1000 IU to 100,000 IU) in children hospitalized with severe pneumonia did not impact the time to resolution, duration of hospitalization, or mortality rate [15]. The small number and poor quality of trial data in the meta-analysis were of concern. Recently published trials were also dealt with small sample sizes and methodological issues [14,16,37,38].

Results are pending from at least ten vitamin D clinical trials on childhood respiratory infections from Northern Australia, India, New Zealand, Bangladesh, Iran, Chile, and South Africa [39]. These studies include a range of outcome measure including recurrent infection,

**Table 3. Risk factors for severe pneumonia (presence of danger signs, hypoxemia, prolonged stay).**

| Characteristics | With danger signs | | Hypoxemia | | Prolonged stay | |
|---|---|---|---|---|---|---|
| | Adjusted OR[1] | p-value | Adjusted OR[1] | p- value | Adjusted OR[1] | p-value |
| Vitamin D level | | | | | | |
| < 75 nmol/L[2] | 1.04 (0.38–2.86) | 0.94 | 1.47 (0.34–6.31) | 0.61 | 0.61 (0.25–1.51) | 0.28 |
| < 50 nmol/L[2] | 0.80 (0.25–2.56) | 0.71 | 1.34 (0.29–6.13) | 0.70 | 0.72 (0.25–2.08) | 0.55 |
| < 25 nmol/L[2] | 1.10 (0.20–5.95) | 0.92 | 6.41 (0.77–53.34) | 0.09 | 1.10 (0.20–5.95) | 0.92 |
| Younger age (2–5 months old) | 1.53 (0.61–3.85) | 0.36 | 0.77 (0.20–3.01) | 0.71 | **2.91 (1.23–6.91)** | **0.02** |
| Younger Gestational age (in weeks) | 1.11 (0.93–1.31) | 0.26 | **1.34 (1.07–1.67)** | **0.01** | 0.99 (0.84–1.16) | 0.88 |
| Low birth weight (< 2500 g) | 1.28 (0.35–4.67) | 0.71 | **4.59 (1.14–18.50)** | **0.03** | 1.09 (0.33–3.68) | 0.89 |
| Increased maternal age | 0.99 (0.92–1.06) | 0.69 | 1.02 (0.93–1.11) | 0.72 | 0.97 (0.91–1.03) | 0.30 |
| Weight for age | | | | | | |
| -2 SD to 2 SD | Ref | | Ref | | Ref | |
| < - 2SD (underweight) | 0.94 (0.27–3.25) | 0.92 | **5.95 (1.54–23.04)** | **0.01** | 1.55 (0.52–4.62) | 0.43 |
| < - 3SD (severely underweight) | 0.93 (0.21–4.16) | 0.92 | **9.58 (1.46–62.34)** | **0.02** | **8.53 (1.89–38.49)** | **0.005** |
| Weight/length for age | | | | | | |
| > 2 SD | 3.14 (0.35–28.56) | 0.31 | n/a | | 1.88 (0.22–16.49) | 0.57 |
| -2 SD to 2 SD | Ref | | Ref | | Ref | |
| < - 2SD (wasted) | 1.81 (0.35–9.42) | 0.48 | 11.26 (0.86–147.47) | 0.07 | 1.38 (0.27–6.94) | 0.70 |
| < - 3SD (severely wasted) | 2.36 (0.29–18.86) | 0.42 | n/a | | 2.89 (0.40–20.75) | 0.29 |
| Early introduction of solid foods[3] | 2.45 (0.86–6.92) | 0.09 | 1.91 (0.51–7.10) | 0.34 | 0.45 (0.15–1.37) | 0.16 |
| Anemia[4] | 1.00 (0.37–2.72) | 0.99 | 2.00 (0.42–9.66) | 0.39 | 0.93 (0.38–2.23) | 0.86 |
| Received antibiotics prior to admission | 0.89 (0.35–2.28) | 0.81 | 0.33 (0.09–1.28) | 0.11 | 0.45 (0.20–1.04) | 0.06 |

[1]Adjusted for gender, passive smoking exposure, socio-economic status, overcrowding house.

[2]Adjusted for ender, passive smoking exposure, socio-economic status, overcrowding house, and exclusive breastfeeding status.

[3] Solids given before 6 months.

[4]Anemia was defined as hemoglobin concentration of < 12.6 gram/dL in age under 6 months, < 12 gram/dL in age between 6 and 23 months, and < 12.5 gram/dL in between 24–59 months.

n/a–there was no sufficient number to run the analysis for the association between exposure and outcome.

duration of stay, admission to ICU, respiratory distress and mortality compared between study participants receiving vitamin D supplementation and placebo [39]. In combination with our study these data will provide an improved understanding of the impact of vitamin D deficiency on the severity of hospitalized childhood pneumonia globally, and the potential role of vitamin D supplementation to reduce pneumonia severity [14]. Larger numbers of trials available for meta-analysis are required to give a more definite answer on the impact of the intervention and inform clinical and public health policy and practice.

We found that young age (<6 months) and low birth weight and poor nutritional status on admission were risk factors for severe pneumonia, consistent with findings from other studies [40,41]. Undernutrition is well known to be associated with the frequency, severity, and mortality of pneumonia due to the weakened host defense mechanisms against infections and decreased respiratory muscles functions [42–45]. Even in vaccinated individuals, malnutrition is associated with a reduction of immunogenicity post-vaccinations [44]. There is a vicious cycle of poor nutritional status and infections [42,43]. While malnutrition decreases innate and adaptive immune functions and increases susceptibility to infections, infections themselves are associated with reduced intake and malabsorption of nutrients due to mucosal injury and altered gut lumens [43]. Low birth weight is also a known risk factor for malnutrition in later childhood [43] as found in our study.

The relationship between vitamin D and inflammation is not completely understood. There have been conflicting data on the role vitamin D deficiency may play on acute and resolving respiratory infection with one study reported that vitamin D status was unlikely to be altered during the acute and recovery phase whereas another study reported that vitamin D deficiency could be a consequence of acute and chronic inflammation [46,47]. It has been hypothesized that vitamin D modulates the host defense mechanism to control the immune response, including the suppression of inflammatory cytokines expressions that can contribute to life-threatening lung inflammation and impaired oxygenation [48,49]. Cathelicidin is an antimicrobial peptide that serve as the direct target gene of vitamin D receptors and is associated with an up-regulation of pathogen clearance [50,51]. However, in a randomized controlled trial in children in India with more than 30% participants had vitamin D level < 37.5 nmol/L, high dose vitamin D administration of 100,000 IU did not confirm significant clinical benefit or confer alteration of serum cathelicidin expression in two week following supplementation [50]. A case control study of 80 Egyptian children demonstrated a positive correlation between serum vitamin D level and cathelicidin levels in case group (mean serum vitamin D 34.87±16.6 nmol/L, r > 0.60) which both serum parameters were measured on the admission day [52]. The timing of vitamin D and cathelicidin measurements in both studies were different which may reflect different cathelicidin expression.

Limitations of our study included the small sample size which may have diminished the power to detect an association between vitamin D deficiency and pneumonia severity. Further, few of our participants presented with danger signs, including hypoxemia. Future studies in the population with a high prevalence of severe pneumonia, i.e., significant proportion of young children with hypoxemia, ICU admission or treatment failure to first line antibiotics, should be considered. We did not investigate the etiology of pneumonia that might have considerable impact on pneumonia severity. Strengths of the study included a recruitment period of over 12 months enabled us to include both seasons, wet and dry. This is important as both respiratory infection and vitamin D are deemed to be seasonal. A standardized of serum vitamin D measurement was employed in our study and we tested different cut off vitamin D levels for association with pneumonia severity in the analysis.

## Conclusion

Vitamin D deficiency was common in under-five children hospitalized with pneumonia but was not related to pneumonia severity and hospitalization outcomes. Data from ongoing clinical trials are needed for better elucidation of the effects of vitamin D supplementation on the severity of pneumonia, particularly in LMIC settings such as Indonesia.

## Supporting information

**S1 File. Study case report form and questionnaire.**
(PDF)

## Acknowledgments

We thank all participants and parents who participated to this study, all study site staffs and pediatricians from district hospitals in Kota Yogyakarta and Kulon Progo who helped with the study recruitment. Lastly, we especially thank to all IPADS research assistants who assisted with data collection and study conduct.

## Author Contributions

**Conceptualization:** Vicka Oktaria, Rina Triasih, Stephen M. Graham, Julie E. Bines, Yati Soenarto, Michael W. Clarke, Margaret Danchin.

**Data curation:** Vicka Oktaria, Mike Lauda.

**Formal analysis:** Vicka Oktaria, Michael W. Clarke.

**Funding acquisition:** Vicka Oktaria, Stephen M. Graham, Yati Soenarto, Margaret Danchin.

**Investigation:** Vicka Oktaria, Rina Triasih, Stephen M. Graham, Julie E. Bines, Yati Soenarto, Margaret Danchin.

**Methodology:** Vicka Oktaria, Rina Triasih, Stephen M. Graham, Julie E. Bines, Margaret Danchin.

**Project administration:** Mike Lauda.

**Resources:** Vicka Oktaria, Stephen M. Graham, Yati Soenarto, Margaret Danchin.

**Supervision:** Vicka Oktaria, Rina Triasih, Stephen M. Graham, Julie E. Bines, Yati Soenarto, Margaret Danchin.

**Validation:** Vicka Oktaria, Stephen M. Graham, Michael W. Clarke, Mike Lauda, Margaret Danchin.

**Writing – original draft:** Vicka Oktaria.

**Writing – review & editing:** Vicka Oktaria, Rina Triasih, Stephen M. Graham, Julie E. Bines, Yati Soenarto, Michael W. Clarke, Mike Lauda, Margaret Danchin.

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
