## [Decision Letter · Decision Letter 0]

12 May 2021

PONE-D-21-03177

Vitamin D deficiency and severity of pneumonia in Indonesian children

PLOS ONE

Dear Dr. Oktaria,

Thank you for submitting your manuscript to PLOS ONE. After careful consideration, we feel that it has merit but does not fully meet PLOS ONE’s publication criteria as it currently stands. Therefore, we invite you to submit a revised version of the manuscript that addresses the points raised during the review process.

We look forward to receiving your revised manuscript.

Kind regards,

Michal Zmijewski

Academic Editor

PLOS ONE

Additional Editor Comments:

Please, consider some additional corrections:.

1. Information concerning vitamin D supplementation is missing.

2. I agree with reviewer one, please recalculate the data using 75 nmol/mL cutoff.

It would be informative to plot at least most impotent clinical data (severity, hypoxemia) against vitamin D concentration

and try to find out if very is any correlation.

Journal Requirements:

2. Please include additional information regarding the parent-interview questionnaire used in the study and ensure that you have provided sufficient details that others could replicate the analyses. For instance, if you developed the questionnaire as part of this study and it is not under a copyright more restrictive than CC-BY, please include a copy, in both the original language and English, as Supporting Information. If the questionnaire is published, please provide a citation to the (1) questionnaire and/or (2) original publication associated with the questionnaire.

Reviewers' comments:

Reviewer's Responses to Questions

**Comments to the Author**

1. Is the manuscript technically sound, and do the data support the conclusions?

Reviewer #1: Yes

2. Has the statistical analysis been performed appropriately and rigorously? 

Reviewer #1: Yes

3. Have the authors made all data underlying the findings in their manuscript fully available?

Reviewer #1: Yes

4. Is the manuscript presented in an intelligible fashion and written in standard English?

Reviewer #1: Yes

5. Review Comments to the Author

Reviewer #1: This manuscript found that vitamin D deficiency was not associated with severity of pneumonia for infants in Indonesia. This finding is in general agreement with other studies.

One limitation of the manuscript is that no detailed information was given on the distribution of 25(OH)D concentrations. Perhaps a histogram could be supplied. Also, the use of 50 nmol/L was not justified. I understand that 50 nmol/L is normally considered the cutoff for vitamin D deficiency. However, there are different cutoffs of 25(OH)D for different health outcomes. The authors should try different cutoffs and find the one that yields the strongest correlation, perhaps also with the lowest 95% CI interval.

the total serum 25-

130 hydroxyvitamin D3 and 25-hydroxyvitamin D2 concentrations were measured using liquid

131 chromatography-tandem mass spectrometry

Comment: Please identify the instrument manufacturer and city, (state) and country.

Additional papers to consider citing as found by searching scholar.google.com with “pneumonia infants vitamin D”. The authors might search there further.

The incidence of acute respiratory infection in Indonesian infants and association with vitamin D deficiency.

Oktaria V, Danchin M, Triasih R, Soenarto Y, Bines JE, Ponsonby AL, Clarke MW, Graham SM.PLoS One. 2021 Mar 23;16(3):e0248722. doi: 10.1371/journal.pone.024872

Effect on the incidence of pneumonia of vitamin D supplementation by quarterly bolus dose to infants in Kabul: a randomised controlled superiority trial.

Manaseki-Holland S, Maroof Z, Bruce J, Mughal MZ, Masher MI, Bhutta ZA, Walraven G, Chandramohan D.Lancet. 2012 Apr 14;379(9824):1419-27. doi: 10.1016/S0140-6736(11)61650-4.

Vitamin D supplementation for severe pneumonia--a randomized controlled trial.

Choudhary N, Gupta P.Indian Pediatr. 2012 Jun;49(6):449-54. doi: 10.1007/s13312-012-0073-x. E

Effect of Vitamin D Supplementation in the Prevention of Recurrent Pneumonia in Under-Five Children.

Singh N, Kamble D, Mahantshetti NS.Indian J Pediatr. 2019 Dec;86(12):1105-1111. doi: 10.1007/s12098-019-03025-z

Effect of Vitamin D Supplementation in the Prevention of Recurrent Pneumonia in Under-Five Children - Correspondence 2.

Kumar P, et al. Indian J Pediatr. 2020. PMID: 32170491 No abstract available.

Effect of Vitamin D Supplementation in the Prevention of Recurrent Pneumonia in Under-Five Children - Correspondence 1.

Kumar J, et al. Indian J Pediatr. 2020. PMID: 32172465 No abstract available.

Vitamin D Status in Neonatal Pulmonary Infections: Relationship to Inflammatory Indicators.

El-Kassas GM, El Wakeel MA, Elabd MA, Kamhawy AH, Atti MA, El-Gaffar SAA, Hanafy SK, Awadallah E.Open Access Maced J Med Sci. 2019 Dec 14;7(23):3970-3974. doi: 10.3889/oamjms.2019.592.

Vitamin D and LL-37 in children with pneumonia

EAM Albanna, YF Ali, RAM Elkashnia - Egyptian Journal of Pediatric …, 2010 - ajol.info

… tract infection (77.0 nmol/l) and hospital controls (77.2 nmol/l), the average vitamin D status of

these individuals was greater than 75 nmol/l, which can be explained by that all the studied infants

consumed vitamin D through fortified infant formula or … Pneumonia …

  Cited by 19 Related articles All 5 versions 

https://www.ajol.info/index.php/ejpai/article/view/108522

Is anything known about the source of the pneumonia, i.e., bacterial, fungal, or viral?

6. PLOS authors have the option to publish the peer review history of their article (what does this mean?). If published, this will include your full peer review and any attached files.

Reviewer #1: No

---

## [Author Response · Author response to Decision Letter 0]

24 Jun 2021

PONE-D-21-03177

Vitamin D deficiency and severity of pneumonia in Indonesian children

Response to the reviewers

Academic editors

1. Information concerning vitamin D supplementation is missing

Authors’ response

We have added more information relevant to vitamin D supplementation trials in the discussion section. 

In our study participants, routine vitamin D supplementation in under five children was not a standard practice during our data collection. We did not collect information on vitamin D supplementation in our participants. 

Change made in the manuscript

There have been few clinical trials on supplementation of vitamin D for severe pneumonia in hospital settings [15]. Oral vitamin D supplementation 1000 - 2000 IU/day for 5 days in Indian children (age 2 to 59 months) hospitalized with pneumonia was not associated with the resolution time from pneumonia as indicated by the disappearance of chest indrawing and/or danger signs, and hours of hospital stay, but the sample size was small (N=20) [34]. On the contrary, a higher single dose of oral vitamin D (100,000 IU) was effective to prevent from pneumonia recurrency within 90 days in Afghan children with non-severe pneumonia (N= 453, age 1 to 36 months) but a similar dose of vitamin D supplementation given quarterly in healthy infants in the same population (N=3,046) did not show prevention from getting pneumonia [35, 36]. (Line 285 - 294, Page 16)

2. I agree with reviewer one, please recalculate the data using 75 nmol/mL cut-off. It would be informative to plot at least most impotent clinical data (severity, hypoxemia) against vitamin D concentration and try to find out if very is any correlation.

Authors’ response

Thank you for highlighting this point. We have performed additional data analysis and tested different level of vitamin D cut-off (< 75 nmol/L, < 50 nmol/L, and < 25 nmol/L). However, the changes of cut offs did not materially change the overall results and conclusion (Table 3, Page 13-14)

Change made in the manuscript

There is no consensus on the recommended serum vitamin D cut off for respiratory health. We defined vitamin D deficiency as a serum (or plasma) 25-hydroxyvitamin D (25(OH)D) concentrations < 50 nmol/L. This is the recommended cut off for optimal bone homeostasis [25]. In this paper, we mainly used < 50 nmol as the cut-off for deficiency but different cut off levels (< 75 nmol/L and < 25 nmol/L) were also tested in our analysis. (Line 136 -141, Page 6)

For additional analysis results, please see changes in Table 3 (Page 13-14). 

3. Please include additional information regarding the parent-interview questionnaire used in the study and ensure that you have provided sufficient details that others could replicate the analyses. For instance, if you developed the questionnaire as part of this study and it is not under a copyright more restrictive than CC-BY, please include a copy, in both the original language and English, as Supporting Information. If the questionnaire is published, please provide a citation to the (1) questionnaire and/or (2) original publication associated with the questionnaire.

Authors’ response

We have attached our study questionnaire and case report forms in two language versions (S1 File)

Change made in the manuscript

Please see S1 File

Reviewer #1: 

1. This manuscript found that vitamin D deficiency was not associated with severity of pneumonia for infants in Indonesia. This finding is in general agreement with other studies. One limitation of the manuscript is that no detailed information was given on the distribution of 25(OH)D concentrations. Perhaps a histogram could be supplied. Also, the use of 50 nmol/L was not justified. I understand that 50 nmol/L is normally considered the cutoff for vitamin D deficiency. However, there are different cutoffs of 25(OH)D for different health outcomes. The authors should try different cutoffs and find the one that yields the strongest correlation, perhaps also with the lowest 95% CI interval.

Authors’ response

We have provided an additional figure (histogram) in the manuscript (Fig 2. The distribution of serum vitamin D level on admission). 

We have also performed additional data analysis and tested different level of vitamin D cut-off (< 75 nmol/L, < 50 nmol/L, and < 25 nmol/L). However, the changes of cut offs did not materially change the overall results and conclusion (Table 3, Page 13-14)

Change made in the manuscript

There is no consensus on the recommended serum vitamin D cut off for respiratory health. We defined vitamin D deficiency as a serum (or plasma) 25-hydroxyvitamin D (25(OH)D) concentrations < 50 nmol/L. This is the recommended cut off for optimal bone homeostasis.(Paxton et al. 2013) In this paper, we mainly used < 50 nmol as the cut-off for deficiency but different cut off levels (< 75 nmol/L and < 25 nmol/L) were also tested in our analysis. (Line 136 - 141, Page 6)

For additional analysis results, please see changes in Table 3 (Page 13-14). 

2. The total serum 25-hydroxyvitamin D3 and 25-hydroxyvitamin D2 concentrations were measured using liquid chromatography-tandem mass spectrometry

Comment: Please identify the instrument manufacturer and city, (state) and country.

Authors’ response

We have added information on instrument manufacturer and city, (state) and country in the manuscript.

 Change made in the manuscript

Sera from venous blood were assayed at Metabolomics Australia, Centre for Microscopy, Characterization, and Analysis, University of Western Australia, Perth. The total serum 25-hydroxyvitamin D3 and 25-hydroxyvitamin D2 concentrations were measured using liquid chromatography-tandem mass spectrometry using an Agilent 6460 2D LC-MS/MS, Singapore. This was the current gold standard for vitamin D measurement, which has shown excellent agreement with CDC’s 25(OH)D3 inaugural vitamin D standardization program (r2 =0.99).[23] The limit of detection and the limit of quantitation of the assay for 25-hydroxyvitamin D3 is 0.5 nM and 2 nmol/L, consecutively. (Line 159 - 166, Page 7)

3. Additional papers to consider citing as found by searching scholar.google.com with “pneumonia infants vitamin D”. The authors might search there further.

a. The incidence of acute respiratory infection in Indonesian infants and association with vitamin D deficiency. Oktaria V, Danchin M, Triasih R, Soenarto Y, Bines JE, Ponsonby AL, Clarke MW, Graham SM.PLoS One. 2021 Mar 23;16(3):e0248722. doi: 10.1371/journal.pone.024872

b. Effect on the incidence of pneumonia of vitamin D supplementation by quarterly bolus dose to infants in Kabul: a randomised controlled superiority trial. Manaseki-Holland S, Maroof Z, Bruce J, Mughal MZ, Masher MI, Bhutta ZA, Walraven G, Chandramohan D.Lancet. 2012 Apr 14;379(9824):1419-27. doi: 10.1016/S0140-6736(11)61650-4.

c. Vitamin D supplementation for severe pneumonia--a randomized controlled trial. Choudhary N, Gupta P.Indian Pediatr. 2012 Jun;49(6):449-54. doi: 10.1007/s13312-012-0073-x. E

d. Effect of Vitamin D Supplementation in the Prevention of Recurrent Pneumonia in Under-Five Children. Singh N, Kamble D, Mahantshetti NS.Indian J Pediatr. 2019 Dec;86(12):1105-1111. doi: 10.1007/s12098-019-03025-z

e. Effect of Vitamin D Supplementation in the Prevention of Recurrent Pneumonia in Under-Five Children - Correspondence 2.

f. Kumar P, et al. Indian J Pediatr. 2020. PMID: 32170491 No abstract available. Effect of Vitamin D Supplementation in the Prevention of Recurrent Pneumonia in Under-Five Children - Correspondence 1.

g. Kumar J, et al. Indian J Pediatr. 2020. PMID: 32172465 No abstract available.

h. Vitamin D Status in Neonatal Pulmonary Infections: Relationship to Inflammatory Indicators. El-Kassas GM, El Wakeel MA, Elabd MA, Kamhawy AH, Atti MA, El-Gaffar SAA, Hanafy SK, Awadallah E.Open Access Maced J Med Sci. 2019 Dec 14;7(23):3970-3974. doi: 10.3889/oamjms.2019.592.

i. Vitamin D and LL-37 in children with pneumonia. EAM Albanna, YF Ali, RAM Elkashnia - Egyptian Journal of Pediatric …, 2010 - ajol.info… tract infection (77.0 nmol/l) and hospital controls (77.2 nmol/l), the average vitamin D status of these individuals was greater than 75 nmol/l, which can be explained by that all the studied infants consumed vitamin D through fortified infant formula or … Pneumonia …

Authors’ response

Thank you very much for the suggestions. We have included some of those articles in the manuscript text as appropriate. Please see the changes throughout the manuscripts. 

Changes in the manuscript

a. Community acquired pneumonia accounted for 18% of all hospitalizations in Indonesian children < 5 years with a case fatality rate between 5% and 20% [9]. Vitamin D deficiency has been reported common in one in every three to five Indonesian children aged between 6 months to 12 years old, and nine in every ten newborns [10, 11]. (Line 72-76, Page 4)

b. Vitamin D deficiency at birth has also been associated with more frequent episodes of acute respiratory infections non pneumonia in the first year of life of Indonesian infants [11]. (Line 80-82, Page 4)

c. There have been few clinical trials on supplementation of vitamin D for severe pneumonia in hospital settings [15]. Oral vitamin D supplementation 1000 - 2000 IU/day for 5 days in Indian children (age 2 to 59 months) hospitalized with pneumonia was not associated with the resolution time from pneumonia as indicated by the disappearance of chest indrawing and/or danger signs, and hours of hospital stay, but the sample size was small (N=20) [34]. On the contrary, a higher single dose of oral vitamin D (100,000 IU) was effective to prevent from pneumonia recurrency within 90 days in Afghan children with non-severe pneumonia (N= 453, age 1 to 36 months) but a similar dose of vitamin D supplementation given quarterly in healthy infants in the same population (N=3,046) did not show prevention from getting pneumonia [35, 36]. Different doses of vitamin D tested, and parameters of pneumonia severity have contributed to inconsistent results between trials, and it is difficult to interpret the effect of vitamin D on pneumonia severity based on the assessment of a singular study. A more comprehensive analysis for multiple studies in a Cochrane meta-analysis study of seven randomized controlled trials (N=1,529 participants) published in 2018, reported that vitamin D supplementation (doses 1000 IU to 100,000 IU) in children hospitalized with severe pneumonia did not impact the time to resolution, duration of hospitalization, or mortality rate [15]. The small number and poor quality of trial data in the meta-analysis were of concern. Recently published trials were also dealt with small sample sizes and methodological issues [14, 16, 37, 38]. (Line 285 - 303, Page 16)

d. However, in a randomized controlled trial in children in India with more than 30% participants had vitamin D level < 37.5 nmol/L, high dose vitamin D administration of 100,000 IU did not confirm significant clinical benefit or confer alteration of serum cathelicidin expression in two week following supplementation [50]. A case control study of 80 Egyptian children demonstrated a positive correlation between serum vitamin D level and cathelicidin levels in case group (mean serum vitamin D 34.87±16.6 nmol/L, r > 0.60) which both serum parameters were measured on the admission day [52]. The timing of vitamin D and cathelicidin measurements in both studies were different which may reflect different cathelicidin expression. (Line 352 - 360, Page 17 - 18)

4. Is anything known about the source of the pneumonia, i.e., bacterial, fungal, or viral?

Authors’ response

Unfortunately, we did not collect information on the pathogens causing pneumonia and it was not a routine procedure in the district hospitals in our settings.

We have stated this in the study limitation “We did not investigate the etiology of pneumonia that might have considerable impact on pneumonia severity”. (Line 367 -368, Page 18)

Change in the manuscript

None

---

## [Decision Letter · Decision Letter 1]

29 Jun 2021

Vitamin D deficiency and severity of pneumonia in Indonesian children

PONE-D-21-03177R1

Dear Dr. Oktaria,

We’re pleased to inform you that your manuscript has been judged scientifically suitable for publication and will be formally accepted for publication once it meets all outstanding technical requirements.

Kind regards,

Michal Zmijewski

Academic Editor

PLOS ONE

Additional Editor Comments (optional):

Thank you for addressing all questions raised by reviewers.

Reviewers' comments:

Reviewer's Responses to Questions

**Comments to the Author**

1. If the authors have adequately addressed your comments raised in a previous round of review and you feel that this manuscript is now acceptable for publication, you may indicate that here to bypass the “Comments to the Author” section, enter your conflict of interest statement in the “Confidential to Editor” section, and submit your "Accept" recommendation.

Reviewer #1: All comments have been addressed

2. Is the manuscript technically sound, and do the data support the conclusions?

Reviewer #1: (No Response)

3. Has the statistical analysis been performed appropriately and rigorously? 

Reviewer #1: (No Response)

4. Have the authors made all data underlying the findings in their manuscript fully available?

Reviewer #1: (No Response)

5. Is the manuscript presented in an intelligible fashion and written in standard English?

Reviewer #1: (No Response)

6. Review Comments to the Author

Reviewer #1: (No Response)

7. PLOS authors have the option to publish the peer review history of their article (what does this mean?). If published, this will include your full peer review and any attached files.

Reviewer #1: No

---

## [Editor Report · Acceptance letter]

1 Jul 2021

PONE-D-21-03177R1 

Vitamin D deficiency and severity of pneumonia in Indonesian children 

Dear Dr. Oktaria:

I'm pleased to inform you that your manuscript has been deemed suitable for publication in PLOS ONE. Congratulations! Your manuscript is now with our production department. 

Kind regards, 

on behalf of

Dr. Michal Zmijewski 

Academic Editor

PLOS ONE